# Effects of Chia Seeds on Growth Performance, Carcass Traits and Fatty Acid Profile of Lamb Meat

**DOI:** 10.3390/ani13061005

**Published:** 2023-03-09

**Authors:** Selene Uribe-Martínez, Juan Antonio Rendón-Huerta, Verónica Guadalupe Hernández-Briones, Alicia Grajales-Lagunes, Juan Ángel Morales-Rueda, Gregorio Álvarez-Fuentes, Juan Carlos García-López

**Affiliations:** 1Coordinación Académica Región Altiplano Oeste, Universidad Autónoma de San Luis Potosí (UASLP), Carretera Salinas-Santo Domingo #200, Salinas 78600, SLP, Mexico; uribemartinezselene@outlook.com; 2Facultad de Ciencias Químicas, Universidad Autónoma de San Luis Potosí (UASLP), San Luis Potosí 78210, SLP, Mexico; veronicaghernandezb@hotmail.com (V.G.H.-B.); grajales@uaslp.mx (A.G.-L.); 3Viscoelabs Materials Research Center, San Luis Potosí 78216, SLP, Mexico; ja.morales@viscoelabs.com; 4Instituto de Investigación de Zonas Desérticas, Universidad Autónoma de San Luis Potosí (UASLP), San Luis Potosí 78377, SLP, Mexico; gregorio.alvarez@uaslp.mx (G.Á.-F.); jcgarcia@uaslp.mx (J.C.G.-L.)

**Keywords:** carcass, linolenic acid, PUFAs, Rambouillet lambs

## Abstract

**Simple Summary:**

Growth is an important factor in animal production. Polyunsaturated fatty acids in human foods have been shown to have health benefits. Dietary manipulation strategies such as the inclusion of oilseeds in diets have been used to enhance the FA profile of sheep meat. Chia seeds are rich in polyunsaturated fatty acids and fiber. This study investigated the effect of increasing the amount of dietary chia seeds on lamb growth performance and changes in the fatty acid profile of the meat. The inclusion of chia seeds in lambs’ diets to feed lambs increased the bodyweight of neither the meat carcasses nor the non-meat components. However, it tended to increase the oleic acid and decrease stearic acid in the meat (*Longissimus thoracis*).

**Abstract:**

The aim of this work was to supplement a diet with chia seeds (*Salvia hispanica* L.) based on the requirements of finishing lambs for meat, and to analyze biometric parameters and fatty acid profiles in meat. Eighteen male Rambouillet lambs with a bodyweight of 25 kg were used. Animals were kept in individual pens with water and feed provided ad libitum. Three finishing diets were designed with the inclusion of 0, 50 and 100 g dry matter chia seeds and divided among the animals (*n* = 6). The experimental period lasted 60 days. The weights of the individual lambs were recorded every 14 days. At the end of the experiment, the animals were slaughtered and the weights of the hot carcasses and non-meat components were registered. In addition, an analysis of the fatty acid composition was carried out in the muscles (*Longissimus thoracis*). The total weight gain and average daily gain displayed significant differences (*p* < 0.05). Initial and final bodyweights, such as the dry matter intake, did not display differences. The fatty acid profile of the meat tended to decrease the SFA (stearic acid) and increase MUFA (oleic acid) (*p* < 0.0001) when chia seeds were added to the lamb diets. In conclusion, chia seed supplementation did not increase meat production or other biometric parameters; however, it modified the fatty acid profile in *L. thoracis*.

## 1. Introduction

In Mexico, the demand for sheep is high, due mainly to the use of its meat in the preparation of barbacoa [1]. Mostly creole sheep are produced, although various pure breeds such as Suffolk, Hampshire, Rambouillet and Corriedale are also produced [2]. French et al. [3] pointed out that it is important to note that red meat contains high levels of saturated fatty acids (SFAs) and low levels of polyunsaturated fatty acids (PUFAs), and that this is also related to the increase in concentrate in diets. Saturated fatty acid synthesis is linked to the extensive biohydrogenation of dietary unsaturated FAs that takes place in the rumen by rumen microbiota; these SFAs are then absorbed and deposited as fat in meat [4].

Currently, red meat is an important dietary component in Western countries and is typically low in n-3 fatty acids and high in SFA and n-6 fatty acids [5]. Although dairy and meat contain high levels of SFAs, they also provide a source of monounsaturated fatty acids (MUFAs) and PUFAs that have beneficial properties for human health [6]. However, it has been evidenced that some SFAs and trans-MUFAs negatively affect blood lipid profiles and are associated with an increased risk of coronary events [7]. To avoid this, the modulation of fatty acid profiles in meat in order to reduce the SFA content can be approached via high stearoyl-CoA desaturase (SCD) activity to increase the synthesis of n-3 PUFAs, which could help to prevent cardiovascular diseases in humans [8]. Consequently, dietary manipulation strategies such as the inclusion of oilseeds in diets have been used to enhance the FA profile of sheep meat for human consumption. Several studies have been conducted with vegetable oils such as linseed, canola, rapeseed and flaxseed rich in n-3 PUFAs to modify the fatty acid profiles of lamb meat [4,7,9,10]. Furthermore, chia seeds also provide a rich source of n-3 PUFAs [11].

Chia seeds (*Salvia hispanica* L.) are native to the central valleys of Mexico and Northern Guatemala; since the Mayan and Aztec ages, chia seeds have been considered an important staple crop used as a foodstuff. These seeds are rich in omega-3 fatty acids (α-linolenic acid), contain soluble and insoluble fibers and proteins, as well as containing minerals and antioxidants [12]. The seeds are rich in lipids (35%) and omega-3 fatty acids, protein (18%) and fiber (23%) [13]. Peiretti and Gai [14] pointed out that chia seed oil is rich in n-3 PUFAs (64.1 g/100 g of total fatty acids) and also contains phenolic compounds, mainly quercetin and kaempferol, which act as strong antioxidants [15]. In previous studies, chia seeds have been supplemented in diets for livestock to increase the omega-3 content and to reduce the risk of cardiovascular diseases. The majority of work in this regard has been performed on monogastric animals such as poultry and rabbits; only a few works have focused on ruminants [16]. Some studies that have used chia seeds in diets for domesticated animals have focused on the modulation of the fatty acid profile. More specifically, chia seeds modify the fatty acid composition of the fat of finishing pigs, where lower proportions of palmitic, stearic and arachidic acids are found with chia seed treatments [5]. On the other hand, Tres et al. [17] mention that not only the fatty acid composition of ingredients but also their oxidative quality (antioxidants) affect the final nutritional and stability properties of meat, liver and plasma in rabbits and chickens. Studies of chia seed supplementation in the diets of ruminants are scarce.

On this basis, under the hypothesis that the inclusion of chia seeds (*S. hispanica*) in the diets of growing lambs modifies the growth performance and fatty acid profile of lamb meat, the objective of this trial was to evaluate the inclusion of chia seeds in the diets of lambs and to analyze its effect on growth performance and the fatty acid profile of lamb meat (*Longissimus thoracis*).

## 2. Materials and Methods

The experiment was carried out in March and April 2021 in a private production unit in Salinas de Hidalgo, San Luis Potosí, Mexico (22°3639″ N and 101°42′45″ W) at 2100 m above sea level; the climate is semiarid with an annual mean temperature of 18.7 °C and receives 391 mm of rainfall. All animal management and care procedures were carried out in accordance with the Mexican Official Standard NOM 062 [18] and in compliance with the regulations established by the Animal Protection Law enacted by the State of San Luis Potosi, Mexico, and they were approved on 28 October 2020 by the Academic Committee CA0CA04CAO-CAI-FMR-05.

### 2.1. Animals

This study used eighteen male Rambouillet lambs of similar live weight (25 ± 2.4 kg; 3 months of age) that were housed in individual cages (1.5 m × 1.0 m) in a ventilated barn. Three TMR diets were designed with 0, 50 and 100 g chia seeds/g DM and were randomly assigned to the animals. At the beginning of the trial, lambs had a 14-day TMR adaptation period with free access to water and feed, which was offered twice a day at 8:00 and 16:00 h. The experiment lasted for 60 days. The feed intake was measured daily, allowing for up to 10% orts. Every 15 days, lambs were weighed before being fed in the morning. Weight gain was calculated by subtracting the initial weight from the final weight. The average daily gain (ADG) was calculated from the changes in bodyweight. The feed conversion ratio was calculated by dividing the dry matter intake (DMI) by ADG.

### 2.2. Feeds and Chemical Composition of Diets

Three finishing total mixed rations (TMRs) for lambs with moderate growing potential were formulated and supplemented with 0, 50 or 100 g of chia seeds per kg of feed (dry matter), with similar contents of crude protein according to the nutritional requirements recommended by the NRC [19] for small ruminants (Table 1) to meet the requirements for finishing lambs between four and seven months of age, weighing 40 kg, with a diet comprising 16% crude protein. The forage to concentrate ratio was 50:50.

Feed samples (500 g) were collected weekly and dried (60 °C for 24 h) for subsequent chemical analysis; the crude protein, ash and acid detergent fiber contents were determined following the AOAC directions [20], and the neutral detergent fiber analysis was modified by adding α-amylase to feed samples [21].

### 2.3. Carcass and Non-Meat Component Evaluation

Once the 60 d feeding period ended, the feed and water were removed for 12 h. Lambs were desensitized by electrocution and slaughtered via exsanguination by slitting their throats in the municipal slaughterhouse of Salinas de Hidalgo, San Luis Potosí, México, under veterinary inspection according to the Mexican Official Standard NOM 033 [22].

Immediately, the weight of the hot carcass and non-meat components (skin, head, heart, liver, lungs, kidney, rumen, trachea, intestines, penis and testicles) were registered on a digital balance. Next, the hot carcass was refrigerated at 4 °C for 24 h and the weight of the cold carcass was recorded [23].

### 2.4. Acid Profiles in Meat

Samples of *Longissimus thoracis* muscle were taken from the 10th to 13th rib [4]. Muscle samples were packed into polyethylene bags and stored at −30 °C until analysis. The fatty acid (FA) profile of the fat extracted from the internal part of the muscle was analyzed by gas chromatography (GC) of the methyl-esters derived from the fatty acids of the samples. The fat methylation process was performed by mixing 10 mg of sample with 0.25 mL of a sodium methoxide solution 5% (*w*/*v*) in methanol and heating it at 60 °C for 30 min [24]. The methyl-esters were analyzed and quantified by GC using a Stabilwax (Restek Corp., Bellefonte, PA, USA) capillary column (60 m × 0.25 mm × 0.25 µm; Varian 3400, Palo Alto, CA, USA). The FA peaks were identified through comparison with known reference methyl-esters (Supelco 37 Component FAME Mix, 47885-U, Sigma-Aldrich Co., Bellefonte, PA, USA).

### 2.5. Statistical Analysis

The data were analyzed through a completely randomized design. Data for the weight of carcasses and non-meat components and fatty acid profiles of meat fat were subjected to a one-way ANOVA with diet as a fixed effect (0, 50 and 100 g dry matter). Six lambs were used per treatment. The data growth performance was analyzed using the ‘mixed’ SAS procedure. The model included the labels lamb (random effect) and residual (lamb within treatment). The average daily weight gain, feed intake and feed conversion were analyzed using the same model. The covariance structure that resulted in the lowest Akaike’s information was heterogeneous autoregressive (ARH) (1). Significant differences were accepted when *p* < 0.05 and were obtained using the statistical software SAS v.9.2 [25].

## 3. Results

Experimental diets had similar nutrient compositions (Table 1). Adding 50 and 100 g DM of chia seeds to diets reduced the steam-flaked corn content by 22% and sorghum content by 50%.

### 3.1. Growth Performance

Lambs fed diets with 0, 50 and 100 g of chia seeds had similar growth performances (Table 2). The initial and final bodyweights did not display differences between treatments (*p* > 0.05). The total weight gain displayed statistical differences in contrast with 0% chia seed inclusion (*p* < 0.05); however, the total weight gain was similar across treatments with chia seed supplementation. The ADG also displayed differences: it was similar in treatments containing 50 and 100 g of chia seeds. The inclusion of chia seeds in the diets did not affect feed intake, but rather, feed intake was similar across the three experimental diets. Although no statistical differences were found in performance measurements, the lambs fed 100 g of chia seeds had the best feed conversion ratio (FCR).

### 3.2. Non-Meat Components of Lambs Supplemented with Chia Seeds

Table 3 shows the weights of hot carcasses and non-meat components. The statistical analysis did not display differences between treatments in the weight of hot carcasses and some of the non-meat components. The major components of the animals’ weight were the carcass, rumen and skin: 18.6, 7.6 and 6.4 kg, respectively.

### 3.3. Fatty Acid Profile in Meat

Table 4 shows the fatty acid profiles (saturated, mono-unsaturated and poly-unsaturated acids (SFAs, MUFAs and PUFAs)) of samples of lamb (*Longissimus dorsi*) meat. Differences were observed in the highest representative proportions of the SFAs (palmitic and stearic acid); stearic acid had a tendency to decrease when the chia seed level increased in the diet (*p* < 0.05). On the other hand, statistical differences were detected within the MUFA group; the biosynthesis of oleic acid represents the highest proportion of MUFAs. Oleic acid had a tendency to increase when the diets were supplemented with chia seeds. In general, as the SFA groups represented the highest proportion of total fatty acids followed by MUFAs and PUFAs, and as the SFA and MUFA groups displayed differences, the SFA content tended to decrease and the MUFA content tended to increase when the diets were supplemented with chia seeds (50 and 100 g). PUFAs were not biosynthesized, as we expected.

## 4. Discussion

Our results are in accordance with those reported by Urrutia et al. [26], who fed lambs with diets supplemented with linseed or chia seeds, both of which are rich in α-linolenic acid (ALA, C18:3n-3); both diets had no effect on the growth parameters. the effects of the inclusion of chia (*Salvia hispanica* L.) seeds in diets on the growth performance of rabbits did not display significant differences among the groups for parameters such as live weight, live weight gain, feed consumption, feed conversion ratio, carcass yield or percentages of edible organs [27].

In our results, ADG differences were observed between diets; the energy intake could have been partially responsible for the ADG differences in favor of a higher chia seed content, given that chia seeds are a good source of energy and protein, and partially replaced sorghum. Although they did not fully comprise energy intake, the chia seeds’ protein contribution was higher when they were in the same proportion as sorghum and even more so when chia seeds were in a higher proportion. This greater amount of chia seed protein may have contributed to the growth of the rumen microbial population and the production of true protein. On the other hand, the tannins contained in sorghum could also inhibit the growth of rumen microorganisms.

Lambs fed with 10% chia seeds or linseed showed a high average daily gain (ADG ≈ 300 g) [28]. These results coincide with those obtained in our trial. Chia seed inclusion did not affect the growth parameters negatively.

Marino et al. [29] reported that the final bodyweight and average daily gain (ADG ≈ 230 g) were not affected in Merino lambs fed a diet containing linseed, quinoa seed or a combination of both (11%). The results of this study agreed with the literature. Regarding rapeseed and linseed oil supplementation in diets for lambs, the bodyweight and average daily gain (ADG ≈ 295 g) were not affected by the type of dietary lipid supplement [5]. Our ADG results for lambs were close to 300 g, which coincide with those obtained in the previously mentioned studies.

An important point for the use of chia seeds in animal nutrition is the absence of off-flavors associated with other popular omega-3-rich oilseeds such as linseed, false flax or fish oils that frequently reduce consumer acceptability [15].

The meat from lambs (Navarra breed) that were fed a diet supplemented with 100 g of chia seeds had increased levels of MUFAs and PUFAs, along with decreased SFA biosynthesis. This response might have been mediated by the regulation of several genes involved in lipogenesis; such regulation seems to be tissue-specific [26]. Jiménez et al. [11] mention that chia seeds have a high proportion of PUFAs (74%) in contrast with other ordinary ingredients used in ruminants’ diets, such as corn, soybean, silages, etc. In our trial, the inclusion of 100 g of DM of chia seeds in the diet is the reason that the animals displayed the highest proportion of MUFAs.

A 10% inclusion of oilseeds (chia or linseed) did not alter the growth rate of Andorra lambs or their carcass quality [28]. In another study, chia seeds greatly improved the fatty acid profile of lamb meat compared with a standard soybean-based control and increased meat α-linolenic acid (ALA) content, raising it from the 0.52 of the control group to 1.73 g/100 g total FA in the chia seed-fed lambs [30]. In our study, ALA was not increased, although the MUFA content increased with the increased chia seed supplementation.

In a trial where lambs were abomasally infused with linseed oil or echium oil through a polyethylene catheter over 4 weeks, the authors concluded that both the linseed oil and echium oil supplementations were similarly effective in enriching lamb meat with long-chain n-3 PUFAs [10].

A study conducted by Schettino et al. [31] in which the diets for dairy goats were supplemented with 2.7 or 5.5% chia seeds, pointed out that the milk yield did not improve, but that the fatty acid profile of the milk was modified and the proportion of short- and medium-chain fatty acids decreased (C10:0, C12:0, C14:0 and C16:0). In the same study, mono- and polyunsaturated fatty acids such as C18:1n-9 cis and C18:2 cis-9 trans-11 increased with respect to the control diet.

The inclusion of chia seeds in the diets of rabbits was effective in improving the n-3 polyunsaturated fatty acid contents of meat and increasing the lipid oxidation in hind-leg meat [32]. The inclusion of chia seed in diets for rabbits increased the PUFA content and decreased the SFA biosynthesis in *L. thoracis* [27], which is similar to the results found in this study. The opposite takes place when diets are supplemented with animal fat, i.e., when supplemented with beef tallow: the higher proportion of SFAs (palmitic and stearic acids mainly) than UFAs (MUFAs and PUFAs) are found in lamb meat (*L. dorsi*) [33].

It should be noted that although chia seeds are high in omega-3 fatty acids, these were not increased in the 5% and 10% treatments. This low level of PUFA biosynthesis in meat could be due to the biohydrogenation process that takes place in the rumen due to gastric isomerase bacteria (*Butyrivibrio fibrisolvens* and *Propionibacterium acnes*), which changes the cis configuration of the unsaturated fatty acids, especially α-linolenic acid, to the trans position, resulting in stearic acid and various intermediates such as rumenic acid (RA, cis-9 trans-11CLA) and vaccenic acid (VA, trans-11C18:1) [34,35,36,37,38]. On the other hand, Manso et al. [38] pointed out that one of the most widely used strategies to increase the levels of VA and RA in meat and milk has been to increase their levels in the rumen using linoleic acid- and α-linolenic acid-rich fats, such as vegetable oils and fats.

Silva et al. [39] conducted an in vitro trial to investigate the effects of chia seeds (CS, 5.5%) compared with flaxseeds (FLAX, 5.0%) and calcium soaps with palm oil fatty acid (MEG, 3.8%) supplementation on the ruminal metabolism in an alfalfa hay-based diet. The authors reported that the dietary treatments had similar amounts of total fat and that the apparent ruminal digestibility, microbial efficiency and N metabolism were similar. However, higher rumen concentrations of C18:0 were observed in FLAX and CS diets, indicating that both the CS and FS fatty acids were highly biohydrogenated. At the same time, both oilseeds increased the concentrations of C18:3 n-3, C20:4 n-6 and total PUFAs.

Other oilseeds used to modify the fatty acid profile in animal production are linseed and sunflower seeds. For instance, when both were supplemented at 6% in grass silage-based diets for dairy goats, they did not affect the nutrient intake, digestibility, milk yield or milk composition. Moreover, both can reduce the palmitic acid and increase the oleic acid [40]. Additionally, in a study where Navarra lambs were fed 14% flaxseed in their diets, the final bodyweight, carcass characteristics and SFA content were not affected compared to a control diet. Nonetheless, despite this, the fact that the PUFA content, mainly C18:3 n-3 (ALA), was significantly increased in intramuscular fat (*Longissimus dorsi*) could be explained by the downregulated gene expression of ACACA, SCD, FADS1 and FADS2 [41].

Finally, perhaps increasing the chia seed supplementation in the diets used in this trial could significantly increase the omega-3 concentration in intramuscular fat.

## 5. Conclusions

Increases in the chia seed inclusion in the diets of growing lambs does not depreciate their growth performance (weight gain, carcass weight). On the contrary, the inclusion of 100 g/kg DM of chia seeds in lamb diets displayed the best results in the fatty acid (FA) profile. Slight changes in the *Longissimus thoracis* FA profile were detected here, which can benefit consumers’ health.

## Figures and Tables

**Table 1 animals-13-01005-t001:** Experimental diets and chemical composition of diets with different amounts of chia seeds.

	Chia Seeds (g/kg DM)		
	0	50	100
Ingredients (g/kg of DM)			
Steam-flaked corn	230	230	180
Chia seeds	0	50	100
Sorghum	100	50	50
Soybean (440 g/kg of CP)	150	150	150
Alfalfa	150	150	150
Corn stubble	350	350	350
Vitamins and minerals premix *	20	20	20
Chemical composition ^Þ^			
Dry matter (g/kg)	889 ± 3.2	891 ± 2.1	893 ± 2.6
Crude protein (g/kg of DM)	154 ± 2.0	158 ± 1.6	163 ± 1.8
Ether extract (g/kg of DM)	27 ± 0.9	39 ± 0.8	50 ± 1.2
Neutral detergent fiber (g/kg of DM)	393 ± 3.1	419 ± 3.5	446 ± 3.8
Acid detergent fiber (g/kg of DM)	247 ± 2.5	260 ± 2.4	272 ± 2.3
Ash (g/kg of DM)	48 ± 0.7	49 ± 0.5	50 ± 0.5
Metabolizable energy (MJ/kg of DM) ^§^	10.7	10.5	10.4

DM = Dry matter. CP = Crude protein. MJ = Mega Joules. (^Þ^ values represent mean ± standard deviation). * Dry matter basis: calcium, 220 mg/kg; phosphorus, 280 mg/kg; magnesium, 0.5%; urea, 102 g/kg; salt, 845 g/kg; vitamin A, 150 MUI/kg; vitamin B, 25 MUI/kg; vitamin E, 150 UI/kg; Sulphur, 30 g/kg; selenium, 10 mg/kg; potassium, 215 mg/kg; iron, 50 mg/kg; cobalt, 20 mg/kg; zinc, 50 mg/kg; manganese, 1600 mg/kg; copper, 300 mg/kg; lasalocid, 1.3 g/kg. ^§^ Calculated values.

**Table 2 animals-13-01005-t002:** Growth performance and feed intake.

Variable	Chia Seed (g/kg of DM)			*p*-Value	SEM
	0	50	100		
Initial bodyweight, kg	25.7	25.3	23.3	n.s.	1.33
Final bodyweight, kg	42.4	45.3	43.2	n.s.	1.23
Total weight gain, kg	16.7 ^b^	20.0 ^a^	19.9 ^a^	*	1.35
Average daily gain, kg	0.28 ^b^	0.33 ^a^	0.33 ^a^	*	0.02
Dry matter intake, kg/d	1.74	1.85	1.78	n.s.	0.23
FCR, kg DM/kg gain	6.2	5.5	5.3	n.s.	0.61

n.s. means not significative, SEM means standard error of the mean, * = *p* < 0.05. Means in the same row with different superscripts are different (*p* < 0.05).

**Table 3 animals-13-01005-t003:** Means of weight of non-meat components of finishing lambs fed diets with chia seeds.

Variable (kg)	Chia Seeds (g/kg of DM)			*p*-Value	SEM
	0	50	100		
Hot carcass	18.4	18.4	18.6	n.s.	0.71
Cold carcass	17.6	17.6	17.8	n.s.	0.67
Heart	0.18	0.17	0.18	n.s.	0.01
Intestines	1.95	1.88	1.90	n.s.	0.04
Liver	0.65	0.62	0.63	n.s.	0.02
Kidneys	0.18	0.20	0.18	n.s.	0.02
Lungs	0.49	0.52	0.48	n.s.	0.02
Rumen	7.39	7.69	7.63	n.s.	0.34
Testicles	0.38	0.34	0.32	n.s.	0.02
Penis	0.10	0.12	0.11	n.s.	0.01
Trachea	0.40	0.43	0.42	n.s.	0.06
Head	1.90	1.78	1.86	n.s.	0.08
Skin	6.83	6.56	6.40	n.s.	0.59

SEM means standard error of the mean, n.s. means not significative.

**Table 4 animals-13-01005-t004:** Effect of addition of chia seeds to the diet on the fatty acid profile (g/100 g of FAME) of lamb meat (*Longissimus thoracis*).

Fatty Acid (g/100 g of Fat)	Chia Seeds (g/kg of DM)			SEM	*p*-Value
	0	50	100		
C14:0 Myristic	1.9	1.9	2.0	0.11	n.s.
C14:1 Myristoleic	0.3	0.5	0.1	0.09	n.s.
C15:0 Pentadecanoic	0.8 ^a^	0.8 ^a^	0.4 ^b^	0.06	**
C16:0 Palmitic	17.2 ^c^	19.8 ^ab^	21.8 ^a^	0.98	*
C16:1 Palmitoleic	2.0	1.3	2.0	0.34	n.s.
C17:0 Margaric	2.5 ^ab^	2.8 ^a^	1.7 ^c^	0.16	**
C17:1 Heptadecenoic	1.9	1.8	1.6	0.20	n.s.
C18:0 Stearic	56.8 ^a^	46.7 ^b^	41.4 ^c^	1.97	***
C18:1 *n:9* Oleic	15.6 ^c^	23.7 ^b^	28.2 ^a^	1.37	***
C18:2 *n:6* Linoleic	0.7 ^a^	0.7 ^a^	0.4 ^b^	0.07	**
C18:3 *n:3* Linolenic	0.1	0.1	0.1	0.02	n.s.
C20:0 Eicosanoid	0.1	0.1	0.2	0.02	n.s.
C20:1 Paullinic	0.0	0.0	0.0	0.00	n.s.
C21:1	0.1	0.0	0.0	0.00	*
∑SFAs	79.3 ^a^	72.1 ^b^	67.5 ^c^	1.48	***
∑MUFAs	20.0 ^c^	27.4 ^b^	32.3 ^a^	1.41	***
∑PUFAs	0.8 ^a^	0.8 ^a^	0.3 ^b^	0.07	*

FAME means fatty acid methyl-esters, SEM means Standard Error of the Mean, n.s. means not significative, * = *p* < 0.01, ** = *p* < 0.001. *** = *p* < 0.0001. Means in the same row with different superscripts are different (*p* < 0.05). SFAs: saturated fatty acids (Σ C14:0, C15:0, C16:0, C17:0, C18:0, C:20); MUFAs: monounsaturated fatty acids (Σ C14:1, C16:1 n-9, C17:1, C18:1 n-9, C20:1, C21:1); PUFAs: polyunsaturated fatty acids (Σ C18:2 n-6, C18:3, n-3).

## Data Availability

The data are available upon request from the corresponding author.

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
