# Peer review of "Effects of Chia Seeds on Growth Performance, Carcass Traits and Fatty Acid Profile of Lamb Meat"

_animals, 2023, doi:10.3390/ani13061005_

Round 1

Author Response

Response to reviewer

The authors thank the helpful comments made by the reviewers, below there are the response to each comment. As requested, each revision or correction was highlighted with the word´s track changes function.

Review article animals-2210756

Title: Effects of chia seeds on growth performance, carcass traits and 2 fatty acids profile of  lamb meat

Reviewer 1.

I suggest making some corrections before publishing.

There is no research hypothesis; please insert in the introduction section.

Answer: Thank you for your advice. It has been added.

Lines 72-74: I suggest to remove the sentence “The inclusion of 15% of chia seeds in rabbit diets improve the content of n-3 PUFA in meat, since it increases the lipid oxidation in the meat of the hind  legs [17]” ; same citation in the Discussion section (Lines 244-246)

Answer: Suggestion was attended.

Materials and methods section:

I suggest reversing paragraphs 2.1 and 2.2 by putting the one on animals first

Answer: Suggestion was attended

Lines 97-98: the authors describe that they collect feed samples weekly to perform chemical analysis; why are the results of these analyzes not presented?

Answer: The table was complete as you suggest, the data is presented in means ± standard deviation.

Line 103: Specify the sex of lambs

Answer: suggestion was attended.

Line 112: explain the acronym DMI

Answer: DMI means Dry Matter Intake, it was explained in the text.

Line 125: How a place of the collection of Longissimus dorsi muscle sample was standardized?

Answer: To determine the fatty acid profile, samples were taken from the internal part of muscle.

Results section:

Table 1: I suggest to shift this table in materials and methods section if you do not present results concerning chemical analyses on feed samples

Answer: The table was completed adding the results in terms of mean ± standard deviation.

Line 156: correct 150 g

Answer: Thank you for your observation. It has been corrected.

Reviewer 2 Report

No comment or reference to energy intake was made.  Were diets alike in Net Energy? Did the inclusion of chia increase energy concentration.  Any comment on differences in ether extract fraction.

Could energy intake had been partially responsible for ADG differences if favor of higher chia content.

Any additional comment in unsaturated fat intake and effect on total DM intake?  Was the level of chia insignificant to affect DMI?

Lines 39 thru 42. Red meat contains high levels of SFA not only because of grain feeding…  Please rewrite concept.

Line 45.  Red meat dietary component of diets… This is redundant.

Line 55.  Replace Diverse for several...

Line 59.  Chia seeds are rich in lipids...  Have to rephase sentence.

Line 73.  Improves  .   Singular.

74. .. chia seeds in the diet of ruminants are scarce.

Longissimus dorsi.  Replace for Longissimus thoracis.

Feed conversion is defined as DMI/ADG. And later the term is defined as kg DM/kg of meat.  These are 2 different relationships.  The study determined Feed to gain ratio or DMI/ADG, but it did not measure meat.  Meat gain?  There is basic terminology confusion here!.  Meat production is equated as similar to live weight gain, but it is not correct.

Line 159  similar in.. replaced for similar for.

162.  Did not affect feed intake..

162.  Similar for the...  Although no statistical differences were found in performance…  This does not match the findings Table 2.  Please revise.

Please, review and rewrite concept of feed conversion

182.  According to statistical analysis...  This is not needed, remove.  Start sentence with:  Differences were…

187,, There is or there was?

213.   Marino et al  … reported..

213 to 214.  Rephrase sentence...

214 to 215,  Results of this study agreed with the literature (not the opposite). Rephrase.

218,  Our results coincided with literature repots (what reports??  Cites) and an ADG….

223 and increased and decreased SFA biosynthesis.  The response might have been mediated…  Such regulation seems to be…   From 223 thru 229. Rephrase.  There are several sentences and concepts in one long sentence.  Need rephrasing for clarity.

235.  increased with the chia level of inclusion.

240  Other studies (cites??)…

245, 247, 248.  Please use past tense and check sentences for mixed use of past and present.

263.  Test to complete the experiment are not matter of the present study. Otherwise, the study is incomplete and requires including these variables. This sentence should be removed.

272.  The study did not determine reduction of cardiovascular disease.  Slight changes in the Longissimus thoracis FA profile were detected here.  Any other conclusion falls beyond the scope of this study.

Author Response

Response to reviewer

The authors thank the helpful comments made by the reviewers, below there are the response to each comment. As requested, each revision or correction was highlighted with the word´s track changes function.

Reviewer 2. Comments and Suggestions for Authors

No comment or reference to energy intake was made.  Were diets alike in Net Energy? Did the inclusion of chia increase energy concentration.  Any comment on differences in ether extract fraction.

Answer. I´m sorry for the omission. The energy was calculated and presented in table 1, as metabolizable energy (ME), values of each ingredient were taken from the Nutritional tables and also taking into account the proportion in the diet. According to the calculations, diets were alike in ME. In the beginning, we tried to formulate the diets with the most isoproteic alike, reducing the amount of corn and sorghum.

Could energy intake had been partially responsible for ADG differences if favor of higher chia content.

Answer. Yes it could, energy intake could have been partially responsible for ADG differences if favor of higher chia content, given that chia seeds are a good source of energy and protein, and partially replaced sorghum. Although they did not fully comprise energy intake, chia seeds´ protein contribution was higher when they were in the same proportion as sorghum and even more so when the chia seeds were in a higher proportion. This grater amount of chia seed protein may contribute to the growth of the rumen microbial population and the production of true protein. On the other hand, the tannins contained in sorghum could also inhibit the growth of rumen microorganisms.

Any additional comment in unsaturated fat intake and effect on total DM intake?  Was the level of chia insignificant to affect DMI?

 Answer. The level of chia could not depress DMI due that the lipids content in the diet is within the limit recommended by the NRC.

Lines 39 thru 42. Red meat contains high levels of SFA not only because of grain feeding…  Please rewrite concept.

Answer. Text was rewritten.

Line 45.  Red meat dietary component of diets… This is redundant.

Answer. We agree, phrase “of diets” was deleted.

Line 55.  Replace Diverse for several...

Answer. Suggestion was attended.

Line 59.  Chia seeds are rich in lipids...  Have to rephase sentence.

Answer. Suggestion was attended.

Line 73.  Improves  .   Singular.

Answer. This part of the text was deleted attending the reviewer 1 suggestion.

  1. .. chia seeds in the diet of ruminants are scarce.

Answer. Suggestion was attended.

Longissimus dorsi.  Replace for Longissimus thoracis.

Answer. Suggestion was attended.

Feed conversion is defined as DMI/ADG. And later the term is defined as kg DM/kg of meat.  These are 2 different relationships.  The study determined Feed to gain ratio or DMI/ADG, but it did not measure meat.  Meat gain?  There is basic terminology confusion here!.  Meat production is equated as similar to live weight gain, but it is not correct.

Answer. I´m sorry for the confusion. Yes, the feed conversion ratio was determined as feed intake/ADG, not meat. It was corrected in the text.

Line 159  similar in.. replaced for similar for.

Answer. Suggestion was attended.

  1. Did not affect feed intake..

Answer. Suggestion was attended.

  1. Similar for the...  Although no statistical differences were found in performance…  This does not match the findings Table 2.  Please revise.

Answer. Text was rewritten to match the findings.

Please, review and rewrite concept of feed conversion

Answer. Observation was attended, it was referred as feed conversion ratio, kg DM/kg of body weight gained.

  1. According to statistical analysis...  This is not needed, remove.  Start sentence with:  Differences were…

Answer. We agree. Phrase was deleted.

187,, There is or there was?

Answer. Thanks for the observation, it was changed to past tense.

  1. Marino et al  … reported..

Answer. Thanks for the observation.

213 to 214.  Rephrase sentence...

Answer. Sentence was rewritten.

214 to 215,  Results of this study agreed with the literature (not the opposite). Rephrase.

Answer. Thank you for the observation

218,  Our results coincided with literature repots (what reports??  Cites) and an ADG….

Answer. Thank you for the observation, the text was rewritten. It refers to the literature previously mentioned. 

223 and increased and decreased SFA biosynthesis.  The response might have been mediated…  Such regulation seems to be…   From 223 thru 229. Rephrase.  There are several sentences and concepts in one long sentence.  Need rephrasing for clarity.

Answer. The paragraph was rewritten.

  1. increased with the chia level of inclusion.

Answer. Thank you for the correction.

240  Other studies (cites??)…

Answer. We refer to the study made by Schettino et al. [30]. This part of the text was rewritten.

245, 247, 248.  Please use past tense and check sentences for mixed use of past and present.

Answer. The text was changed to past tense.

  1. Test to complete the experiment are not matter of the present study. Otherwise, the study is incomplete and requires including these variables. This sentence should be removed.

Answer. The sentence was removed as the reviewer suggested.

  1. The study did not determine reduction of cardiovascular disease.  Slight changes in the Longissimus thoracis FA profile were detected here.  Any other conclusion falls beyond the scope of this study.

Answer. Conclusions were corrected following the reviewer suggestions. We do not have another conclusion.

Attached you will find the file with the changes
